# Endoplasmic Reticulum Stress and Cancer: Could Unfolded Protein Response Be a Druggable Target for Cancer Therapy?

**DOI:** 10.3390/ijms24021566

**Published:** 2023-01-13

**Authors:** Gregorio Bonsignore, Simona Martinotti, Elia Ranzato

**Affiliations:** DiSIT-Dipartimento di Scienze e Innovazione Tecnologica, University of Piemonte Orientale, Viale Teresa Michel 11, 15121 Alessandria, Italy

**Keywords:** ER stress, GRP78, natural compounds, UPR

## Abstract

Unfolded protein response (UPR) is an adaptive response which is used for re-establishing protein homeostasis, and it is triggered by endoplasmic reticulum (ER) stress. Specific ER proteins mediate UPR activation, after dissociation from chaperone Glucose-Regulated Protein 78 (GRP78). UPR can decrease ER stress, producing an ER adaptive response, block UPR if ER homeostasis is restored, or regulate apoptosis. Some tumour types are linked to ER protein folding machinery disturbance, highlighting how UPR plays a pivotal role in cancer cells to keep malignancy and drug resistance. In this review, we focus on some molecules that have been revealed to target ER stress demonstrating as UPR could be a new target in cancer treatment.

## 1. ER Stress and the UPR

The ER plays an important role in eukaryotic cells, as organelle which influence proteins in their synthesis, maturation and folding [1]. ER executes and regulates many of protein post-translational modifications guarantying a correct protein working [2,3].

An optimal protein folding is related to the disulfate-bond formation, that are influenced by several factors, for example ATP, calcium, and oxidizing environment [4]. To guarantee a correct protein folding preventing the accumulation of misfolded or unfolded proteins, ER possesses the ERQC mechanism (ER Quality Control Compartment) [5,6].

ER can be distinguished into two categories: rough and smooth [7]. RER (rough endoplasmic reticulum) is dotted with ribosomes and it is crucial for production, correct folding, quality check and delivery of protein [7]. 

SER (smooth endoplasmic reticulum) is not dotted with ribosomes, and it is paired with smooth slippery fats [6]. Into the SER take place both the production and the metabolism of steroid hormones and fats [8].

The protein quality control is a critical step for the cell survival. ER controls proteins and, in case of misfolding or unfolding, it mediates the ERAD (Endoplasmic-reticulum-associated protein degradation) [9,10,11]. If proteins are unable to pass ERQC, ERAD identifies and destructs them by a proteolytic system [9,12]. Unfortunately, sometimes ERQC functions can be impaired, and this condition causes the development of different serious protein folding diseases, such as neurodegenerative diseases, cardiac diseases, and cancer [13,14].

The first cell component which is involved into signal transduction and homeostatic changes sensing is the ER, giving feedback to other organelles [15]. Typically, proteins are folded (tertiary or quaternary structure) in the ER [1]. However, alterations of ATP level, calcium concentration or redox conditions limit the protein-folding ability of the ER, and as a consequence, there is an aggregation and accumulation of unfolded proteins, determining ER stress [16]. 

UPR is an adaptive response which is used for re-establishing protein homeostasis, and it is triggered by ER stress [17,18]. Three ER proteins mediates UPR activation, i.e., IRE1, PERK, and ATF6. These proteins have a luminal domain binding an ER chaperone called GRP78 which is inactive in normal conditions. In case of ER stress, the luminal domains dissociate from GRP78, causing their activation [19]. UPR mainly plays three roles: -reducing ER stress and re-establishing ER homeostasis → adaptive response;-blocking UPR when ER homeostasis is restored → feedback control;-regulating apoptosis [20,21].

## 2. UPR Pathways

### 2.1. IRE1

IRE1 (Inositol-requiring enzyme 1) is a transmembrane protein of ER (type I), with serine/threonine kinase activity, detecting ER stress through its N-terminal luminal domain, starting also the most common UPR signalling pathway [22]. IRE1 has two isoforms: IRE1α and IRE1β. 

IRE1α is the most studied and is present in all eukaryotic cells. In case of unfolded proteins accumulation, IRE1 oligomerizes in the ER lumen and it starts the autophosphorylation [23]. 

After the activation, IRE1 cuts XBP1 mRNA, causing a shift in its codon reading frame, this condition triggers the formation of a new C-terminal domain which includes an active domain of transactivation, sXBP1 [4,5,24,25].

sXBP1, in turn, provokes the upregulation of UPR-related genes implicated in protein folding, and in translocation to the ER and ERAD [26,27]. 

IRE1 enlists TRAF2, and triggers ASK1 [28]. ASK1 causes the activation of JNK and p38, a MAPK [28,29]. 

Then, JNK molecules move to the membrane of mitochondria leading to Bim activation and the inhibition of Bcl-2. On the other hand, the phosphorylation of p38 MAPK provokes the activation of a transcriptional factor CHOP, causing the increasing of Bim and DR5 expression, at the same time decreasing Bcl-2 expression, this condition leads to the apoptosis initiation [30,31]. 

Bax and Bak can bind to IRE1 and trigger it, interact with IP_3_R inducing the release of Ca^2+^ from the ER [32]. 

### 2.2. PERK

The attenuation of mRNA translation is induced by an ER-resident transmembrane protein called PERK (protein kinase R-like endoplasmic reticulum kinase), which works as sensor of ER stress and contains a luminal domain similar to IRE1 [33,34].

Usually, PERK is linked to GRP78, and, after its activation, it blocks the entrance of newly synthesized proteins into the ER (which is already stressed). After the estrangement from GRP78, PERK forms a dimer and induces its autophosphorylation and activation [35].

This condition is possible for the inactivation of elF2 by the phosphorylation of serine 51 [13]. The inhibition of elF2α is caused by a guanine nucleotide exchange factor complex that leads elF2 to its active GTP-bound form [36], this last reduces the excess of misfolded proteins and mitigates ER stress [11]. 

Moreover, elF2 phosphorylation plays another role; in fact, it allows the translation of UPR-dependent genes, for example ATF4, codifying for different upstream open reading frames [37,38]. ATF4 triggers the expression of ER stress target genes (like CHOP, growth arrest, GADD34 and ATF3) [39,40]. 

### 2.3. ATF6

ATF6 (activating transcription factor 6) is a transmembrane protein of ER (type II), in case of ER stress conditions, dissociates from GRP78 (the release from GRP78 enables ATF6) and moves to the Golgi apparatus for additional proteolytic processing [41,42]. Into the Golgi, two enzymes called site proteases-1 and 2 (S1P-S2P) make the proteolytic cleavage of the full length ATF6 (90 kDa) [43,44]. 

Then, the cleaved N-terminal cytosolic domain of 50-kDa bZIP (cytosolic basic leucine zipper) moves to the nucleus binding to CRE (the ATF/cAMP response elements) and ERSE-1 to induce the transcription of some target protein like GRP78, XBP-1, and CHOP [45,46]. Thereby, during a prolonged ER stress, CHOP can be activated by IRE1, PERK, and ATF6, and then it leads to apoptosis [47,48].

The cleaved ATF6 translocate into the nucleus and works as an active transcription factor to upregulate proteins that improve ER folding capacity (for example two chaperons like GRP78 and GR94), and folding enzymes (for example PDI) [45,49,50,51].

### 2.4. GRP78

GRP78, also known as BiP, is a member of the HSP70 family. It is localized on the membrane of ER of all eukaryotes [52]. GRP78 is composed of 654 amino acids. It edits the folding and assembly, and also avoids the transport of protein (or subunits of them) that are misfolded [53,54,55]. The expression of GRP78 is increased in case of ER stress. For example, GRP78 is upregulated during 

-sugar abrogation;-inhibition of protein glycosylation induced by particular reagents;-intercellular calcium storage disturbance [56].

GRP78 is soluble in water, with only small hydrophobic parts; however, these last components are essential for its function, for example to recognize the unfolded proteins addressed either to the degradation or refolding mechanisms [57]. GRP78 possesses two domains called ABD (or NBD) locates at the N-terminal and SBD at the C-terminal [58]. 

There is 60% homology between GRP78 and the HSP70 family, and in particular, the most conserved domains are ABD and SBD. The most conserved sequence of the HSP70 family belongs to ABD domain [52,58,59]. However, GRP78 differs in protein expression regulation, nevertheless it is an HSP70 family A member, and owns the properties of abnormal protein binding in case of stress. In addition, the protein synthesis inhibitor cycloheximide is effective on GRP78 [60,61] (cycloheximide can inhibit protein synthesis in eukaryotic cells) [62]. 

The state of ER protein folding is checked by three ER-localized transmembrane UPR signal sensors, such as PERK, IRE1, and the transcription factor ATF6.

Under homeostatic conditions, these UPR sensors are maintained inactive by the interaction of their luminal domains with GRP78. 

In response to ER stress, accumulated unfolded proteins sequester GRP78 from the UPR sensors, which promotes the activation of IRE1, PERK, and ATF6 and the induction of GRP78 [25,63]. 

### 2.5. UPR and Apoptosis

UPR acts through the transitional (at transcriptional and transcriptional level) attenuation of the enzymes folding, the induction of ER chaperones, and ERAD involved proteins to relieve protein aggregation in the ER as an adaptive response. 

In case of important and extended ER stress, the UPR triggers some apoptotic pathways [50]. ER stress induces conformational change in ER membrane through the proapoptotic BH3 protein; moreover, it allows Ca^2+^ transmigration to the cytosol, which provokes the activation of m-calpain and, subsequently, the cleavage and the activation of procaspase 12 and caspase cascade [64,65,66,67]. 

CHOP, one of the principal UPR downstream effectors, inhibits Bcl-2, provoking growth arrest and the activation of GADD34 and ERO1, and all these elements promote apoptosis [68,69]. 

The upregulation of GADD34 by CHOP causes a feedback inhibition of elF2α phosphorylation; as a consequence, in cell death and survival, the role of CHOP may be context dependent. This phenomenon could allow the restoration of translation, which could be positive; however, if translation continues also in ER-stress conditions, the accumulation of anomalous proteins may compromise the ER folding ability, causing the death of cells. 

After IRE1 activation, JNK is bound by IRE1 which recruits TRAF2; this condition causes both the release of the procaspase 12 from the ER and the activation of apoptosis signal-regulating kinase 1 and JNK [28,70]. Additionally, PUMA and NOXA are activated by ER stress, and this condition provokes BAX and BAK activation and apoptosis [71]. 

Activation and maintenance of representative UPR pathways in cells treated with low concentrations of chemical ER stress inducers, showed that survival, in case of stress, is reached through intrinsic mRNA instabilities and proteins that induce apoptosis [72]. 

Autophagy is a catabolic process, leading to several effects, in particular, the degradation and recycling of cytosolic, ageing, or misfolded proteins and excess or faulty organelles. ER stress provokes autophagy and promotes cell survival; in fact, in case of starving conditions, it enables the intracellular resources utilization [73]. 

In cells where GRP78 is reduced by small interfering RNA (siRNA), the ER structure is compromised, and autophagosome formation under ER-stress or starvation is suppressed [74].

### 2.6. ER Stress/Calcium-Mediated Apoptosis

The decrease of ER Ca^2+^ causes protein misfolding and chronic mitochondrial Ca^2+^ overload, which lead to apoptosis through Bcl-2-dependent pathway [75]. The localization and oligomerization of pro-apoptotic Bcl-2 proteins, Bax and Bak, are triggered by ER stress, promoting Ca^2+^ release from the ER into the cytosol [50], through IP_3_Rs and RyRs [67,76], which are linked to the apoptotic signal transduction mechanism [77,78,79]. 

When [Ca^2+^]_cyt_ increases, it leads to the activation of Ca^2+^-dependent cysteine protease m-calpain, which is involved in several intracellular processes, for example apoptosis, cell cycle progression, differentiation, and signal transduction [80,81].

m-calpain is well-known to cleave and trigger the ER-resident procaspase-12 [64,82], involved in the ER stress-induced cell death pathway in differentiated PC12 cells [83]. The activation of caspase-12 leads to the activation of procaspase-9 and then to the activation of caspase-3 apoptotic mechanism [65].

The increasing of cytosolic Ca^2+^ causes its uptake into the mitochondrial matrix, inducing a depolarization of the inner mitochondrial membrane and a perturbation of the outer membrane permeability [35]. This leads to the cytochrome c release and to the activation of the apoptosome by Apaf-1, causing apoptosis [84]. 

The most important actor in the regulation of the ER stress-induced apoptosis is CHOP [85]. CHOP is a basic leucine zipper-containing transcription factor, suppressing the expression of Bcl-2 and triggering the transcription of many genes favouring apoptosis [69,86]. The release of procaspase-12 from TRAF2 (and then its activation) is triggered by the association of IRE1/TRAF2 and ER stress [87,88]. The activated caspase-12 results in apoptosis activation [70].

### 2.7. MAM and ER Stress

The ER is closely associated with mitochondria through mitochondria associated membranes (MAM), leading to a strong functional interaction between these two organelles [89]. For example, the acute ER stress impairs mitochondrial function in rat and mouse hearts, while inhibition of mitochondrial respiration by using rotenone or antimycin A also increases the ER stress [90].

ER is a key site of intracellular calcium storage, and ER stress leads to intracellular calcium overload by disrupting calcium homeostasis [91]. ER stress-mediated calcium overload contributes to mitochondrial damage. Furthermore, in response to Ca^2+^ overload, the ER stress also rises ROS production directly through NADPH oxidase 4, and then through impairment of mitochondrial electron transport [92].

Taken together, these data point out the existence of a positive feedback loop between mitochondrial dysfunction and ER stress.

Moreover, the activation of initial phase of UPR, i.e., the pathways to solve the ER stress by expanding the ER, upregulating chaperones and by causing a (temporary) translation stop, is accompanied by ER morphology changes and the consolidation of ER and mitochondria contact sites. 

In fact, considering that the folding of newly proteins is one of most energy-requiring processes, this strengthening of the contact sites makes sense. In ER stress-induced by tunicamycin, an important number of mitochondria relocate towards the perinuclear ER. These mitochondria exhibit a transmembrane potential increase, Ca^2+^ uptake rise, higher ATP production, and increased oxygen consumption [93]. Therefore, tightening the ER-mitochondrial contacts could favour the increase in intracellular ATP level necessary to sustain the pro-survival ER machinery.

Conversely, if ER stress is too severe and cannot be resolved by the UPR, the signalling pathways initiated by the ER stress sensors will turn on a lethal signal, ultimately causing cell death, typically in an apoptotic way [94]. Several studies have also validated a direct relationship between changes in MAM components and deregulated Ca^2+^ transfer and apoptotic sensitivity during ER stress [95,96].

Likewise, new data suggested that the MAM could be involved not only in Ca^2+^ signals, but also in some toxic lipid oxidation products to the mitochondria [97]. This phenomenon could be very important as lipid peroxides may be responsible for spreading ROS signalling, to control mitochondrial Ca^2+^ uptake and cell death choice after ER stress [98].

Additionally, to increase importance of MAM involvement in ER stress, recent studies displayed that ER-mitochondria contacts may be crucial in the formation of autophagosomes [99].

## 3. UPR as Therapeutic Target

One of tumorigenesis hallmark is the unrestrained growth of the transformed cells; cancers are continuously challenged by a limited oxygen and nutrients supply due to inadequate vascularisation. Moreover, some hematopoietic cancers often show that secretory proteins have increased production, such as multiple myeloma cells producing immunoglobulins [100]. All these circumstances induce an ER stress and UPR activation. 

Different cancer types are linked to ER protein folding machinery disturbance, demonstrating as the correct folding process is a key for signalling pathway proteins [101]. A huge amount of evidence demonstrates how the UPR is a crucial process which is very important for cancer cells to keep malignancy and drug resistance. 

UPR is a very active research area because different components are triggered or repressed in malignant cancers [102]. Recent indications suggest that UPR molecular components could be useful as prognostic and diagnostic markers for cancer progression and response to chemotherapeutics [94].

In a great variety of cancers, UPR pathways are activated, and they are essential for tumour microenvironment creation and maintenance. In fact, in cancer cells a great number of events can take place, such as:-over-expression of XBP1;-activation of ATF6;-phosphorylation of eIF2α;-induction of ATF4 and CHOP-upregulation of GRP78;-upregulation of glucose-regulated protein 94 (GRP94, also known as gp96 or HSP90b1);-upregulation of GRP170 [103,104].

From animal studies, it was discovered that XBP1 is important for tumour growth in vivo. In fact, in Xbp1-/- and Xbp1-knockdown mice cells, cancers cannot develop [105]. In addition, ER stress can cause anti-apoptotic reactions. After XBP1 activation, glycogen GSK3b [106] induces p53 phosphorylation, which causes an increasing of its degradation and avoids p53 dependent apoptosis for cancer cells. Furthermore, during ER stress, NFκB is activated and induces anti-apoptotic responses [107]. 

An important role is played by heat shock proteins; in fact, they help cancer cell adaptation against stress associated with oncogenesis both to repair damaged proteins (protein refolding) and to degrade them. Moreover, heat shock proteins are involved in cell proliferation and in resistance to different anti-tumoral drugs that cause apoptosis. For example, HSP90 interfaces with different key proteins in inducing prostate cancer progression, comprised both wild-type and mutated AR, HER2, ErbB2, Src, Abl, Raf, and Akt [108,109]. 

In a great variety of tumours, GRP78 is largely expressed at high levels conferring resistance against therapies in both proliferating and dormant cancer cells. Animal models with reduced GRP78 level shown significant impediments in tumour growth. 

GRP78 mediated-cancer progression has been explained though three major mechanisms: -activation of cancer proliferation;-suppression of apoptosis;-stimulation of angiogenesis [110,111].

ER stress has been involved in different levels of tumour development. The current idea is that, during early tumorigenesis and before angiogenesis occurs, activation of the UPR causes the cell cycle arrest in G1 phase and the activation of p38, both of which induce a dormant state. 

ER stress also promotes anti-apoptotic NF-κB and silences p53-dependent apoptotic signals. If the balance of early cancer development takes off against cell death, ER stress can further induce the aggressive growth of cancer cells by enhancing their angiogenic ability. For example, the induction of GRP170 (which is a BiP-like protein that acts as a chaperone for VEGF [112]) causes an increasing of VEGF secretion. 

Emerging data indicate that agents disturbing UPR pathway may be utilized as promising anticancer drug. However, due to the dual role of the UPR in cell survival/death, and depending on the type of cancer, molecules that either provoke severe ER stress and cell death or compounds blocking the pro-survival role of disturbed UPR of tumour cells could be used either alone or in combination with conventional anticancer treatments [113]. 

### 3.1. GRP78 as Target

In different kinds of cancer, the expression of GRP78 is often elevated if compared to healthy tissues. This has been noticed in different diseases such as hepatocellular carcinoma [114], gliomas [115], prostate [116], and gastric cancers [117]. However, a study on lung cancer showed a link between the overexpression of GRP78 and better prognosis [118]. Despite this, GRP78 results overexpressed in more cases, and, for this reason, the consensus is that its expression is linked to poor prognosis and strong tumour aggressiveness. 

For example, it is well-known that an important expression of GRP78 is correlated with poor prognosis and lymph node metastasis in gastric tumours [117]; moreover, these data are reinforced by preclinical studies which show how the silencing of GRP78 reduces invasion in vitro and tumour growth and metastasis in vivo [117]. 

In prostate tumours, high GRP78 activity is linked to a reduction of patient survival [116], while, in breast tumours [119,120], a shorter time to recidivism is associated with high GRP78 expression [120].

In addition, a GRP78 overexpression has been described both in tumour with acquired anti-oestrogen resistance and in oestrogen-receptor positive breast cancer cells [119]. However, a study reported a correlation between oestrogen receptor positivity and GRP78 or XBP1 expression, showing that the oestrogen upgrade can promote the expression of GRP78 and XBP1 [121].

Moreover, GRP78 might contribute significantly to therapy resistance. In glioma cells and breast cancer cells, the inhibition of GRP78 expression improves sensitivity to chemotherapy [115,122]; in particular, in resistant breast cancer cells, it re-establishes sensitivity to anti-oestrogens [119]. On the other hand, GRP78 overexpression induces resistance to chemotherapeutics [114,115,121,123] and anti-oestrogens [114,119]. 

GRP78-mediated therapy resistance is not a general rule; in renal carcinoma cells, GR78 knockdown causes resistance to chemotherapy [124].

Nevertheless, GRP78 is a possible target for cancer therapies. Chen et al. described the use of the GRP78-promoter to induce the expression of an apoptotic pathway [125,126]. Another strategy is the direct attack against GRP78 expressing cells. For example, EGCG ((−)-epigallocatechin-3-gallate), a green tea polyphenol, can block GRP78 binding on its ATP-binding domain. A treatment with EGCG can damage glioma cells resistance to temozolomide [115]. EGCG is also effective to reduce the proliferation of mesothelioma cell lines, overexpressing GRP78 [127].

An alternative is versipelostatin, a transcriptional inhibitor of GRP78, which causes selective death of glucose-deprived cells and stops cancer development in vivo [128]. XPB1 and ATF4 expression are repressed by the treatment with this compound, but only during glucose deprivation, not after treatment with tunicamycin or A23187. Combining versipelostatin with cisplatin induces growth inhibition [129].

The evidence indicates that GRP78 plays different roles in cancer cells, not only in the ER [129]. In fact, some isoforms of GRP78 have been detected in the nucleus, mitochondria, cytosol, and also in the ER membrane [123,130,131,132,133]. 

Expression of GRP78 on the cell surface could be a great therapeutic target using specific therapeutic antibodies. An example is PAT-SM6, an autoantibody isolated from a gastric cancer patient [133]. PAT-SM6 is directed against a particular isoform of GRP78, showing the ability to inhibit the development of gastric carcinoma in vivo [134]. 

Moreover, this antibody can induce specific apoptosis in multiple myeloma cells, while it is safe against non-malignant cells [135]. A phase I clinical trial shown that PAT-SM6 could also be effective against melanoma [136]. 

### 3.2. PERK and IRE1 as Targets

A comparison between malignant tissue and normal tissue shows different expression of ATF4, the downstream factor of PERK [137]. In neoplastic tissues, the expression of ATF4 is higher than in normal tissues; so, this factor plays a crucial role in the response against chemotherapy. In addition, expression of ATF4 correlates with cisplatin resistance in lung cancer cell lines [138]. In the case of an ATF4 overexpression, it could induce multidrug resistance against cisplatin, doxorubicin, etoposide, irinotecan, and vincristine, but not to 5-fluorouracil [138,139]. The resistance to cisplatin causes an intracellular glutathione level increase [139]. On the contrary, ATF4 knockout cells show a decreased glutathione biosynthesis and higher sensitivity to anti-cancer treatments. 

However, the UPR is helpful not only to improve chemotherapy; in fact, it can also have a significant impact on the radiotherapy efficacy. In breast cancer, radiotherapy triggers the PERK-pathway of the UPR, and this is induced by increased PERK protein levels, phosphorylated eIF2α, ATF4, and LAMP3 [140]. Moreover, in vivo studies on rat intestinal epithelial cells show how irradiation promotes three important consequences:-GRP78 expression;-eIF2α phosphorylation;-XBP1 splicing [141].

According to experimental data, this stress response is not linked to ATF6 [141]. Only during knockdown of PERK, ATF4 or LAMP3 can enhance the sensitivity of breast cancer cells to radiotherapy [140]. Furthermore, a knockdown of GADD34, an essential element to prolong eIF2α phosphorylation, causes radio-resistance; on the contrary, a treatment with a pharmacological PERK inhibitor (GSK2606414) provokes radio-sensitization [140]. 

Another PERK inhibitor (GSK2656157) shows the ability to suppress the development of xenografted tumours through the reduction of the vascular density [142,143]. Tamoxifen is a well-known anti-oestrogen drug which can induce UPR components, such as GRP78 and LAMP3 [121,140]. 

In addition to radio-sensitization of breast cancer cells, knockdown of LAMP3 sensitizes cells to tamoxifen [140]. In breast tamoxifen-tolerant cancer, cells have been noted an increased expression of LAMP3 which, if silenced, provokes a decreasing of tamoxifen-resistance [140]. 

Another arm of UPR is upregulated in breast cancer cells: XBP1-pathway [144]. In fact, in this kind of tumour, XBP1 is co-expressed with the oestrogen receptor. XBP1 overexpression causes the independent growth of oestrogen receptor positive cells regardless of the presence of the hormones. Furthermore, this condition makes cells more resistant to the anti-oestrogen drugs such as tamoxifen and faslodex [144]. According to data, higher levels of spliced XBP1 are linked to more aggressive breast tumours and poor prognosis [145].

### 3.3. ER Stress-Inducing Agents as Anti-Cancer Therapies

There are different molecules which can induce ER stress and some of them have the potential to be used as anti-cancer therapies. For example, eeyarestatin 1 is a small molecule which induces ER stress by preventing ER-associated degradation [146]. Eeyarestatin 1 shows a synergistic behaviour with bortezomib and is selective against cancer cells. Furthermore, tunicamycin-induced ER stress can increase sensitivity of some cancers to therapies, for example: breast cancer cells sensitivity to radiotherapy [147] and ovarian cancer cells sensitivity to cisplatin and carboplatin [148]. 

Other data have shown different situations where ER-stress can induce resistance to chemotherapy [149,150]. Tunicamycin, through induction of GRP78, can significantly reduce the apoptosis induced by chemotherapy [151] and, as a consequence, GRP78 silencing can reduce the sensitivity to tunicamycin effects.

Thus, depending on the treatment given, pharmacological induction of ER stress can be effective both for promoting sensitivity and inducing resistance to anti-tumoral therapies.

Another important study led by Ledoux et al. shows that via the UPR, glucose withdrawal induces the expression of P-glycoprotein in hepatic cancer cells [152], enhancing the efflux of chemotherapeutic drugs. 

### 3.4. UPR-Induced Autophagy Helps to Survive ER Stress

Some conditions (hypoxia and detachment from the extracellular matrix) and compounds (A23187, tunicamycin, thapsigargin and brefeldin A) which cause ER-stress can induce both the UPR and the autophagy [153,154]. 

Autophagy is an important way to reduce ER stress and to allow survival of cells [73,153,155] but this effect was discovered only in cancer cells, not in normal cells [153]. 

As shown by Kouroku et al., polyglutamine aggregates induce ER stress leading to autophagy as a degradation mechanism [156]. In this case, the activation of autophagy is provoked by the PERK-arm of the UPR. Dominant-negative PERK and mutated non-phosphorylatable eIF2α avoid the conversion of LC3-I to LC3-II, while the phosphorylation of eIF2α induces ATG12 expression [156]. 

In addition, in neoplastic conditions, a resistance against therapies is provoked by ER stress-induced autophagy. 

Pharmacological-induced ER stress, before the cisplatin therapy, can induce autophagy and confer resistance against cisplatin-induced apoptosis [157], whereas in breast cancer, PERK-dependent autophagy is induced by radiotherapy [147,158]. It is possible to sensitize cells to radiotherapy though pharmacological inhibition of autophagy or PERK-pathway silencing [140,158]. 

Moreover, autophagy can be induced also by tamoxifen treatment, and this process is mediated by ATF4-induced LAMP3 [159,160] or by GRP78-dependent inhibition of mTOR [119]. Another possible treatment of breast cancer is bortezomib, that leads to an ATF4-dependent increase in LC3B and autophagy, favouring bortezomib-resistance [161]. 

Other reports have indicated that not the PERK-arm, but the IRE1-arm is responsible for UPR-mediated autophagy. In neuroblastoma, ER stress caused by amino acid starvation, thapsigargin, or tunicamycin promotes autophagy [73]. However, this condition can be blocked by IRE1 silencing or a pharmacological treatment with a JNK inhibitor. 

Interestingly, cells lacking PERK or ATF6 induce autophagy similar to wild-type cells, suggesting how autophagy is independently triggered. 

UPR-independent pathway can also induce autophagy in response to ER stress. ER stress caused by thapsigargin or tunicamycin triggers autophagy via protein kinase Cθ (PKCθ) [162]. PKCθ activation takes place independently from UPR sensors. ER stress-activated autophagy can be blocked by PKCθ silencing or its pharmacological inhibition but, this condition can be prevented by chelating intracellular Ca^2+^, while the kinase is not reactive to amino acid starvation. 

Furthermore, when cytosolic Ca^2+^ levels increase, AMPK is triggered by CAMKK-β, this condition leads to ER-stress which causes the inhibition of mTOR [162], inducing autophagy.

## 4. Compounds Targeting the UPR

As stated, ER is an important organelle which controls protein folding, calcium reserve, and lipid production. New proteins move to the ER for other adjustments, such as glycosylating, folding, and disulphide bond formation [163]. To guarantee the correct maturation and folding, ER uses the ERQC mechanism. 

Usually, proteins are inspected by ERQC, or they are degraded through by ERAD [164]. When cells are affected by bad stimuli, such as calcium disruption, glucose deprivation and redox imbalance, ER is full of unfolded or misfolded proteins, triggering ER stress. 

UPR is linked to a correction of folding and also to a decreasing of the proteins amount into ER and gene translation [165], but, if the stress is out of control, cells could trigger the apoptotic mechanism [166]. In traditional medicine, natural compounds played an important role in the treatment of different diseases. In recent times, many of these methods were rediscovered. In particular, some natural molecules have been revealed to target ER stress, which has a central role in the pathophysiological progress of diseases (see Table 1). For example, thapsigargin, produced by *Thapsia garganica*, is deemed an ER stress inducer which uses competitive inhibition of SERCA. 

### 4.1. Natural Compounds and ER Stress-Related Apoptosis

#### 4.1.1. PERK-eIF2α-CHOP

Baicalein (5,6,7-trihydroxyflavone), a flavone originally isolated from the roots of *Scutellaria baicalensis* and *Scutellaria lateriflora* may trigger ER stress, regulating CHOP, JNK (upregulation) and Bcl-2 family (downregulation), thus, it can induce apoptosis and autophagy in hepatocellular carcinoma cells (HCC) [168]. 

Berberine, a quaternary ammonium salt from plants in the genus Berberis, has been employed against several diseases, such as dyslipidaemia, hyperglycaemia, and obesity. It is an antidiabetic agent which can reduce the phosphorylation levels of PERK, eIF2α, and IRS-1-ser307, inducing the reduction of ER stress to increase insulin sensitization in Hep G2 cells [169]. Moreover, berberine enhanced GRP78 by suppression of ubiquitination/proteasomal degradation of GRP78 and activation of ATF6, suggesting as berberine can induces autophagic death via enhancing GRP78 levels [170].

A natural triterpenoid called celastrol, contained into *Tripterygium wilfordii* Hook, can increase the effect of BH3 mimetic drug ABT-737. This combination activates the eIF2α-ATF4 signalling pathway, then upregulates Noxa expression, which can induce apoptosis in HCC cells [171]. 

Another important natural compound is curcumin. In acute promyelocytic leukaemia (APL), it can inhibit ERAD and protease-regulated degradation and then induce the amassing of phosphorylated misfolded N-CoR. This condition causes the sensitization of APL to UPR-induced apoptosis [172]. 

Honokiol, a compound separated from *Magnolia officinalis*, may attenuate ER stress caused by apoptosis in torsion/detorsion testicular injury, it downregulates the expression of p-eIF2α and CHOP [173]. Honokiol treatment can also induce apoptotic pathway in human chondrosarcoma cell lines but not primary cells, triggering ER stress, as shown by changes in Ca^2+^ levels [174].

From *Astragalus membranaceus* a natural compound (astragaloside IV) has been separated, showing protective activity in diabetic nephropathy because of ER stress inhibition [175]. Specifically, it can downregulate PERK-ATF4-CHOP signalling reducing podocyte apoptosis in diabetic rats [176]. Moreover, astragaloside IV sensitized non-small cell lung cancer cells to cisplatin through suppressing ER stress and autophagy [177]

Another natural compound, resveratrol, a polyphenol contained into different fruits and in wine, induces the reduction of fat and body weight trough the regulation of lipid and glucose metabolisms. Resveratrol can downregulate AMPK signalling pathway and activate ER stress through the expression of eIF2α phosphorylation and CHOP [178]. 

#### 4.1.2. GRP78 and IRE1-XBP1

Metformin was originally developed from natural compounds found in the plant *Galega officinalis* (also known as goat’s rue), a traditional herbal medicine in Europe. The mechanism of action of metformin involves an activation of AMP-activated protein kinase (AMPK). The activation of AMPK decreases cell injury during oxidative stress in part through the inhibition of mitochondrial permeability transition pore (MPTP) opening. Moreover, in addition to its anti-diabetic effect, a number of studies suggest that AMPK activators might exert an anti-cancer effect through the modulation of the UPR in ER stress conditions. Interestingly, the effects of metformin on the UPR have recently been described in different cancer cell lines [179,180]. Metformin co-treatment with bortezomib suppresses the induction of GRP78, impairing the autophagosome formation in myeloma cells and enhancing apoptosis of cancer cells [181].

EGCG, a compound in green tea, is an inhibitor of GRP78. It can act both as an antioxidant and as a pro-oxidant. ECGC, due to GRP78 inhibition, may greatly improve the therapeutic effect of temozolomide to cause glioblastoma apoptosis [182] from in vivo studies, for its effect on GRP78, EGCG could protect against cisplatin-induced nephrotoxicity by suppression of ER stress-mediated apoptosis in mouse renal tubular epithelial cells [183]. In malignant mesothelioma, EGCG can induce synergistic effects with gemcitabine and ascorbic acid [184].

Another molecule is nicotine, the main component of tobacco. Nicotine can diminish apoptosis induced by tunicamycin-mediated ER stress in PC12 cells, provoking a reduction of expression for ATF6, GRP78, and IRE1-XBP1 [185]. 

Furthermore, resveratrol possesses two isomers (trans and cis). Both isomers show antioxidant, anti-inflammatory, antitumor, and immunomodulatory properties. In particular, cis-resveratrol may suppress the expression of GRP78 and reduce the production of ROS in human macrophages [186]; in addition, it may abrogate the pro-survival IRE1-XBP1 signalling and activate the pro-apoptotic responses activating ER stress-induced apoptosis [187]. 

### 4.2. Natural Compounds and Calcium-Mediated ER Stress

In many *Curcuma* species, there are three principal constituents for curcuminoids: bisdemethoxycurcumin (BDMC), demethoxycurcumin (DMC), and curcumin. BDMC has shown, in different studies, the ability to induce arrest at S phase and subsequent apoptosis in human lung cancer NCI H460 cells, acting through elevation of Ca^2+^ and ROS and activation of ER stress; in fact, these events are preceded by GRP78, IRE1α, IRE1β, CHOP, ATF6α, ATF6β, and caspase-4 upregulation [188]. In acute myeloid leukaemia, curcumin can cooperate with carnosic acid to provoke apoptosis, the combination is cancer-selective cytotoxic inducing the disruption of Ca^2+^ homeostasis. 

Baicalein promotes apoptosis in retina ganglion cells (N18). It causes MDA-MB-231 cell apoptosis through stimulation of ER stress, reduction of Bcl-2 expression, elevation of Ca^2+^, downregulation of mitochondrial membrane potential and upregulation of Bax expression [168]. 

Another interesting natural compound is camphene, a molecule isolated from the essential oil of *Piper cernuum*. It can turn on apoptosis in melanoma cells, thus inducing ER stress, which could be linked to calcium perturbation and mitochondria disorder [189]. 

A triterpenoid saponins isolated from *Gynostemma pentaphyllum*, called gypenosides shown its ability to activate apoptosis in human hepatoma cells inducing calcium-modulated ER stress and mitochondrial disorder [190].

### 4.3. Natural Compounds and Inflammation-Mediated ER Stress

The anti-inflammatory activity of natural compounds is an important subject of study. In fact, there is a link between ER stress and the inflammatory response. New data have related the UPR induction with various pro-inflammatory factors release such as IL-6, IL-8, and TNF-α [191]. In fact, some UPR signals can facilitate processes which lead to different inflammatory phenomena related to cancer progression [94]. In particular, the maintenance of an inflammatory microenvironment is an essential component of all tumours [94].

For example, curcumin exhibit anti-inflammatory activity against bacterial invasion Caco-2 cells and T84 cells (intestinal epithelial cells), downregulating ER stress and decreasing the expression of GRP78 and IRE1α-XBP1 signalling [192]. 

Another example is the sterol ergosta-7,22-dien-3-ol separated from the echinoderm *Marthasterias glacialis*. It shows anti-inflammatory functions, as confirmed by the upregulation of COX-2, iNOS, IL-6, and NF-κB. It is well-known that CHOP-modulated ER stress upregulates the inflammatory responses, however it can be attenuated by sterol ergosta-7,22-dien-3-ol [193]. 

The TXNIP/NLRP3 inflammasome results ER stress-associated and causes flogosis and cell death in the endothelial disorder. Some natural molecules such as EGCG, quercetin, and luteolin shown the ability to turn on AMPK signalling, this condition leads to the suppression of ER stress and TXNIP/NLRP3 inflammasome [194]. Additionally, curcumin, ilexgenin A, and astragaloside IV can suppress TXNIP/NLRP3 inflammasome through their interaction with AMPK signalling [195,196,197].

### 4.4. Metal (Ruthenium and Iridium) Based Compounds

BOLD-100 (ruthenium complex sodium trans-[tetrachlorido-bis(1Hindazole) ruthenate(III)] (BOLD-100/KP1339) is an inhibitor of stress-induced GRP78 upregulation, disrupting endoplasmic reticulum (ER) homeostasis and inducing ER stress and UPR.

BOLD-100 is linked to UPR response and is the most studied non-platinum metal-based anticancer drugs [198]. It displayed promising anticancer effects in several tumour models; in addition, clinical trials prove its safety [198]. 

Several cancer cells lines as well as tumour models highlighted that BOLD-100 induces a down-modulation of the GRP78 and an induction of ER stress [198]. Moreover, ruthenium possesses an important redox activity which allows it to interfere with the cellular redox balance via direct as well as indirect mechanisms [199]. Ruthenium compounds can play a role in Fenton-like reactions, leading to generation of reactive oxygen species (ROS), and can also induce depletion of the intracellular GSH pools, which induce an increased susceptibility in cells to endogenous and exogenous oxidative stress. As a consequence of this (redox) stress, treatment with ruthenium induces apoptosis of cancer cells via the mitochondrial pathway [200,201]. 

Recently, BOLD-100 obtained encouraging results in the treatment of some non-responsive tumours like breast, gastric, colorectal, and pancreatic cancers.

In malignant mesothelioma cell lines, treatment with BOLD-100 induce an increase in ROS production and Ca^2+^ release from the ER, producing an activation leading to ER stress and, ultimately, to cell death [202].

Some authors have also described as iridium complexes may show effects on ER status. In particular, it has been suggested [203,204] that Ir complex–peptide hybrids (IPHs) induce ER stress and decrease the mitochondrial membrane potential, thus triggering intracellular signalling pathways and resulting in cytoplasmic vacuolization in Jurkat cells.

## 5. Conclusions

The role of UPR in cancer development and progression has been demonstrated; however, we do not recognize exactly the finest regulations and implications of UPR signalling in tumour cells. 

UPR was initially considered to be pro-survival adaptive machinery designed to reduce levels of unfolded proteins and to restore ER homeostasis. This UPR implication may be true for cancer cells at least in the first stages of developments, while UPR could also be utilized in other ways to benefit cancer in advancement and invasion.

Some approaches, as well as some molecules, have demonstrated that UPR could be a druggable target to treat tumour, but the next challenge will be to correctly decipher the UPR signalling details and intricacies to specifically target cancer cells lines and select patients to benefit from this therapeutic option. 

## Figures and Tables

**Table 1 ijms-24-01566-t001:** Natural products affecting ER stress and UPR pathways.

Natural Product	Common Sources	Mechanisms of Action	Chemical Structure Depiction
Baicalein	*Scutellaria baicalensis* and *Scutellaria lateriflora*	Activation of ER stress inducing apoptosis and autophagy	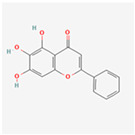
Berberine	*Rhizoma coptidis*	Effective against several diseases, like dyslipidaemia, hyperglycaemia, and obesity.	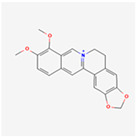
Celastrol	*Tripterygium wilfordii* Hook	Increasing of BH3 mimetic drug ABT-737 effect.	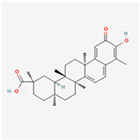
Curcumin	*Curcuma* species	Stimulation the UPR-induced apoptosis.	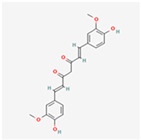
Honokiol	*Magnolia officinalis*	Decreasing of p-eIF2α and CHOP expression.	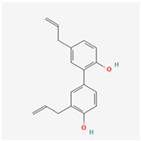
Astragaloside IV	*Astragalus membranaceus*	Protective activity in diabetic nephropathy because of ER stress inhibition.	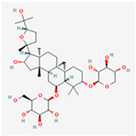
Resveratrol	Almost 70 plant species	Reduction of fat and body weight trough the regulation of lipid and glucose metabolisms.	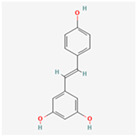
EGCG	*Camellia sinensis*	Inhibitor of GRP78 [167].	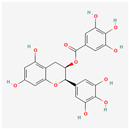
Nicotine	*Nicotiana tabacum*	Downregulation of apoptosis induced by tunicamycin-mediated ER stress in cancer cells (causing the reduction of expression for ATF6, GRP78, and IRE1-XBP).	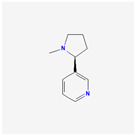
Camphene	*Piper cernuum*	turning on of apoptosis in melanoma cells inducing ER stress.	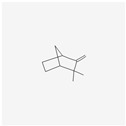
Gypenosides	*Gynostemma pentaphyllum*	Ability to activate apoptosis in human hepatoma cells inducing calcium-modulated ER stress and mitochondrial disorder.	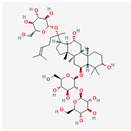

## Data Availability

Not applicable.

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
