# Peer review of "Endoplasmic Reticulum Stress and Cancer: Could Unfolded Protein Response Be a Druggable Target for Cancer Therapy?"

_ijms, 2023, doi:10.3390/ijms24021566_

Round 1
Reviewer 1 Report
This Review by Gregorio Bonsignore et al. analyses the role of the Unfolded Protein Response, following Endoplasmic Reticulum stress, in cancer. More importantly, it highlights the therapeutic opportunities deriving by targeting the Unfolded Protein Response in cancer. Thus, the authors describe a number of compounds able to target different components of the Unfolded Protein Response. The review is exhaustive and references are appropriate. However, in my opinion, a number of points should be addressed.
Major points:
1) The acronimes should be explained, when cited for the fist time, as follows: ERQC (Endoplasmic Reticulum Quality Control).
2) The review appears too schematic. I suggest to the authors to semplify the first paragraph renaming it just "ER stress and the UPR" and avoiding further organisation in subparagraphs.
3) I suggest to the authors to follow a more logical scheme to avoid redundancy. For example, in the paragraph 2 (UPR pathways) I would start the description with the three sensors and then GRP78.
4) Also paragraph 3 and 4 are redundant: 3. UPR as therapeutic target; 4. Targeting ER stress and UPR. I would rename the paragraph 4 "Compounds targeting the UPR".
5) In the last decade a great number of studies have highlighted the anticancer action of metformin, a traditional antidiabetic compound extracted by Galega officinalis. Several of this studies have pointed out that metformin and its derivatives can target the UPR in myeloma and different types of other cancer such as gastric, breast, lung and endometrial cancer. Thus, I would suggest to add a paragraph on metformin and its derivatives.
Minor points:
- page 2 lane 61: the first subparagraph should be 2.1.
- page 4 lane 156: JIK, is that what the authors meant?
- the punctuation use is often absent or inappropriate.
- english is poor and should be improved.
- Typos
Author Response
This Review by Gregorio Bonsignore et al. analyses the role of the Unfolded Protein Response, following Endoplasmic Reticulum stress, in cancer. More importantly, it highlights the therapeutic opportunities deriving by targeting the Unfolded Protein Response in cancer. Thus, the authors describe a number of compounds able to target different components of the Unfolded Protein Response. The review is exhaustive and references are appropriate.
We thank the reviewer for the positive evaluation of our ms.
However, in my opinion, a number of points should be addressed.
Major points:
- The acronimes should be explained, when cited for the fist time, as follows: ERQC (Endoplasmic Reticulum Quality Control).
We have correctly explained the acronyms at the first time
- The review appears too schematic. I suggest to the authors to simplify the first paragraph renaming it just "ER stress and the UPR" and avoiding further organization in subparagraphs.
We have modified according to suggestion.
- I suggest to the authors to follow a more logical scheme to avoid redundancy. For example, in the paragraph 2 (UPR pathways) I would start the description with the three sensors and then GRP78.
We have modified the paragraph 2 according to reviewer comments. In particular, the paragraph now starts with the three sensors and then with GRP78.
- Also paragraph 3 and 4 are redundant: 3. UPR as therapeutic target; 4. Targeting ER stress and UPR. I would rename the paragraph 4 "Compounds targeting the UPR".
We have modified the name of paragraph 4 as suggested.
- In the last decade a great number of studies have highlighted the anticancer action of metformin, a traditional antidiabetic compound extracted by Galega officinalis. Several of this studies have pointed out that metformin and its derivatives can target the UPR in myeloma and different types of other cancer such as gastric, breast, lung and endometrial cancer. Thus, I would suggest to add a paragraph on metformin and its derivatives.
We have inserted information about the effects of metformin in cancer cells.
Minor points:
- page 2 lane 61: the first subparagraph should be 2.1.
- page 4 lane 156: JIK, is that what the authors meant?
- the punctuation use is often absent or inappropriate.
- english is poor and should be improved.
- Typos
We have corrected the mistakes and typos and improved the English
Reviewer 2 Report
The review article reported by Elia Ranzato and the research team, entitled "ER stress and cancer: could UPR be a druggable target for cancer therapy?" The authors focus on some natural compounds that have been revealed to target ER stress demonstrating as UPR could be a new target in cancer therapy. This review is interesting for the development of new molecules for anticancer applications by target ER. I have noticed the following minor lacunae in the review which need to be addressed before accepting this manuscript for publication in the International Journal of Molecular Sciences:
Comments:
1. In the title, the authors should include the full form of ER and UPR.
2. In the abstract, the authors should include the full form of GRP78.
3. Authors should be explained abbreviations, where it first appears, several places missing this.
4. 4.1. Natural compounds and ER Stress-Related Apoptosis, in this section authors include the chemical structure of those natural compounds discussed hereby.
5. In the abstract, the authors wrote this statement "we focus on some natural molecules that have been revealed to target ER stress demonstrating as UPR could be a new target in cancer treatment" but in this review some Ru based compounds targeting ER, authors should change this statement.
6. These are a few papers (Eur. J. Inorg. Chem. 2021, 1796–1814, Molecules 2021, 26, 7028), related to ER targeting metal complexes, authors should in cite them in this review.
7. The title "4.5. Ruthenium based compounds" may change to "4.5. Metal (Ruthenium and Iridium) based compounds".
Author Response
The review article reported by Elia Ranzato and the research team, entitled "ER stress and cancer: could UPR be a druggable target for cancer therapy?" The authors focus on some natural compounds that have been revealed to target ER stress demonstrating as UPR could be a new target in cancer therapy. This review is interesting for the development of new molecules for anticancer applications by target ER.
We thank the reviewer for the positive evaluation of our ms.
I have noticed the following minor lacunae in the review which need to be addressed before accepting this manuscript for publication in the International Journal of Molecular Sciences:
Comments:
- In the title, the authors should include the full form of ER and UPR.
We have modified the title.
- In the abstract, the authors should include the full form of GRP78.
We have modified it.
- Authors should be explained abbreviations, where it first appears, several places missing this.
We have correctly explained the acronimes at the first time
- 4.1. Natural compounds and ER Stress-Related Apoptosis, in this section authors include the chemical structure of those natural compounds discussed hereby.
In Table 1, we have inserted a chemical structure depiction
- In the abstract, the authors wrote this statement "we focus on some natural molecules that have been revealed to target ER stress demonstrating as UPR could be a new target in cancer treatment" but in this review some Ru based compounds targeting ER, authors should change this statement.
We have modified the statement.
- These are a few papers (Eur. J. Inorg. Chem. 2021, 1796–1814, Molecules 2021, 26, 7028), related to ER targeting metal complexes, authors should in cite them in this review.
We have discussed it in the 4.5 section
- The title "4.5. Ruthenium based compounds" may change to "4.5. Metal (Ruthenium and Iridium) based compounds".
We have modified it.
Round 2
Reviewer 1 Report
The authors answered satisfactorily to the points raised by the reviewer